# Adaptation and Validation of the Lithuanian-Language Version of the Olympic Value Scale (OVS)

**DOI:** 10.3390/ijerph19074360

**Published:** 2022-04-05

**Authors:** Saulius Sukys, Daiva Majauskiene, Diana Karanauskiene, Ilona Tilindiene

**Affiliations:** Department of Physical and Social Education, Lithuanian Sports University, Sporto 6, LT-44221 Kaunas, Lithuania; daiva.majauskiene@vdu.lt (D.M.); diana.karanauskiene@lsu.lt (D.K.); ilona.tilindiene@lsu.lt (I.T.)

**Keywords:** Olympic movement, Olympic Values, Lithuanian adaptation, reliability, validity

## Abstract

Background: The Olympic Games is one of the biggest sports events which should express and promote Olympic ideals. Aiming to generate more insights on the expression of Olympic Values, the Olympic Value Scale (OVS) was developed to assess how people perceive values in relation to the Olympic Games. The aim of the present study was to examine the validity and reliability of the Lithuanian version of OVS (LT-OVS). Methods: The scale construct validity and reliability was tested using a sample of 365 university students (mean age 22.02, SD = 6.58; 49.9% male). After the evaluation of the scale structure, convergent and discriminant validity as well as reliability of the scale were evaluated by assessing composite reliability and average variance extracted (AVE), examining the square root of the AVE. For further validity analysis, associations between the LT-OVS factors and other variables were examined. Results: The original OVS captures three dimensions, which are appreciation of diversity, friendly relations with others, and achievement in competition. Exploratory and confirmatory factor analyses confirmed the original three-factor structure of the OVS. The internal consistency values for all three subscales of the LT-OVS were 0.80 and higher. Convergent and discriminant validity criterions were met. Relations between the LT-OVS dimensions and attitudes towards fair play and Olympic Games were also revealed and discussed. Conclusions: This study makes a contribution by confirming the validity of the LT-OVS and encouraging future adaptation of it into other cultures and research on Olympic Values.

## 1. Introduction

The Olympic movement seeks to “contribute to building a peaceful and better world by educating youth through sport practiced in accordance with Olympism and its values” [1] (p. 15) and its culmination is the “bringing together of the world’s athletes at the great sports festival, the Olympic Games” [1] (p. 11). However, the Olympic Games is not just a sports event or a set of different sports competitions [2]; it is a complex of various processes that involve the political, economic, and cultural elite in one whole [3,4]. Today, the Olympic Games have become a mega sports event that not only provides opportunities for the best athletes to compete but also allows them to show national pride and serves as a kind of soft power demonstration tool [5] related to the expression of modern diplomacy [6]. It is no coincidence that countries or cities are competing to host the Olympic Games.

The Olympic Games as a mega sports event receive considerable attention from researchers who seek to understand how the Games are treated and what their perceived social, environmental, and economic effects are. Research shows that before the Olympics, people can expect positive impacts of the event and have more positive attitudes towards the Games [7,8]; however, there may also be inverse expectations when there is more concern about the impact even before the event [9]. People were found to be more positive about the social than the environmental impacts of the Olympic Games [10]. Examining how attitudes change, it was observed that people were more aware of the environmental, societal, and cultural benefits of the Olympic Games than the economic ones [11,12,13]. Additionally, the evaluation of the Olympics and their impact also depends on people’s socio-demographics [14,15], place of residence [8,11], their involvement in decision-making processes [15], or individual values [16]. It should be mentioned that this mega event also receives significant attention in the media, where the positive aspects mostly result from the tendency to present sport and sociocultural legacies [17].

Of course, all of this data needs to be evaluated in the context of the specific Olympic Games. However, it is important to note that perceptions of the positive impacts of the Olympic Games are related to more positive overall attitudes towards the Olympics [15,18]. However, research on the potential or actual impact of the ongoing or past Olympic Games on the local community, the environment, financial resources, and the country itself is still more prevalent. In part, this allows us to judge the population’s attitudes towards the Olympics, but they are sometimes limited to more positive or negative statements about the event [16]. However, such evaluations often do not reflect the extent to which the core values and ideals associated with the Olympic Games are observed, whereas the official agenda of the Olympics calls for the promotion of solidarity and inclusion [1]. Furthermore, the evaluations should be based on the fundamental principles of Olympism, which describe sport as an activity “promoting peaceful society concerned with the preservation of human dignity” [1] (p. 11), and “practising sport, without discrimination of any kind and in the Olympic spirit, which requires mutual understanding with a spirit of friendship, solidarity and fair play” [1] (p. 11). Thus, the Olympic movement seems to be based on values, so it is important to better understand what values people associate with the Olympic Games in their evaluations of them. 

How the values associated with the Olympics are perceived is relevant for several reasons. It can be argued that the Olympic movement itself faces challenges because the International Olympic Committee (IOC) as a global sports organization has been criticized for corruption, lack of accountability, questionable ethics, and social justice problems [19]. In view of the high (multi-million-dollar) cost of the Olympics, support for the organization of the Olympic Games is declining in some countries [13,20]. At the same time, mistrust of sports organizations can also have a negative effect on the evaluation of the values they promote [21]. It is worth noting that the study which revealed that even the Youth Olympic Games, as a sports event, were not able to improve adolescents’ perceptions of the Olympic Values [21]. People’s perceptions of the Olympic Values are also important because most previous studies have focused on how residents of the countries hosting the Olympic Games have supported and perceived the benefits of this sports event, but values related to the Olympic Games were less analyzed, especially in the general population. Such research would provide a better understanding of which values associated with the Olympic Games people believe in more, as well as whether the stakeholders behind the Olympic Movement (both IOC and National Olympic Committee) act according to these values [22]. 

Some recent studies have provided support for the relationship between the perceived values in relation to the Olympic Games and the attitudes and support of those hosting the Olympic Games [23]. Other studies also give reason to believe that people from different countries do not see the Olympics in the same way [2]. It is no coincidence that the relationship between Olympism and nationalism is also discussed [24] as well as the extent to which this may relate to people’s attitudes towards Olympics and Olympic Values [16]. This encourages further research to gain a better understanding not only of people’s attitudes towards the values associated with the Olympic Games, but also whether and how these attitudes differ around the world, both in terms of cultural differences and in terms of changes in the Olympic movement itself.

In emphasizing the importance of the aforementioned research, attention should be paid to previous attempts to analyze Olympic Values. Some studies have sought to interpret how Olympic Values were perceived by athletes [25] and Olympians [26], while others have assessed attitudes toward these values [27] or values in relation to the Olympic Movement [28]. However, some previous studies were qualitative [25,26], and in some other studies, the research instruments that were used lacked information on their validity [27]. Therefore, their results are generally difficult to compare. It should also be noted that, to date, we still lack a valid and reliable scale to capture the Olympic Values and especially values related to the Olympic Games. 

A significant step was taken by Koenigstorfe and Preuss [22], who recently developed a scale that measures Olympic Games-related values (OVS). The OVS is a 12-item self-report free-factor instrument. Specifically, those three factors capture three dimensions: appreciation of diversity (including anti-discrimination, diversity, equality, and tolerance), friendly relations with others (including brotherhood, friendship, understanding, and warm relations with others), and achievement in competition (including achievement, achievement of one’s personal best, competition, and effort). It was found that the OVS positively correlated with individuals’ identification with the national Olympic Game athletes, attitudes towards the Olympic Games, and behavioral intention to attend the Olympic Games [22]. Additionally, this scale has been replicated for Germany [22] and later the Portuguese version was validated as well [23]. This gives reason to believe that the scale is replicable for different cultural contexts. Accordingly, knowledge of the perceptions of the values associated with the Olympic Games by people from different cultures can be useful for stakeholders in the development and implementation of specific programs to effectively communicate the Olympic ideas to people. However, when evaluating valid research instruments related to both Olympic Values and values specifically related to the Olympic Games, we observe that nothing is currently available in the Lithuanian language. Therefore, the aim of this study was to adapt and validate the Lithuanian version of the OVS and in doing so, encourage future research and offer the possibility of including Lithuanian studies on Olympic Values in the international context. 

Several steps have been taken to adapt and validate the OVS. First, translation of the OVS was conducted. Second, we examined the structure of the OVS by conducting an exploratory (EFA) and confirmatory (CFA) factor analysis. Next, several procedures were performed to determine the validity of the data—specifically, this study involved the analysis of convergence validity (we measured composite reliability and average variance extracted (AVE)) and discriminant validity (determined by the square root of the AVE value). The study also included variables related with Olympic Games and fair play for validation purposes. Specifically, based on previous studies [22,23], we expected to find a positive relationship between perceived values related with the Olympic Games and interest in these Games. Different authors include fair play among Olympic Values [21,28], which is completely natural because fair play is also mentioned among the fundamental principles of Olympism, and the IOC’s mission is to encourage fair play [1]. Hence, we expected that those with higher scores on perceived values related with the Olympic Games would demonstrate more positive attitudes towards fair play. 

## 2. Materials and Methods

### 2.1. Participants

The targeted study population for verifying the validation and reliability of the Lithuanian version of OVS was university students, aged 18–31 years, residing in Lithuania. The participants were able to communicate in Lithuanian and complete the questionnaire. There are various recommendations regarding the sample size for factor analysis. According to the criteria for validation studies, some authors [29] offered 100 participants as a minimum while others [30] offered 200 as a fair number and 300 as good. Sample size can be reduced to below 100 with consistently high communalities (when all greater than 0.60) and well determined factors [31]. Some rules also recommended from 5 to 10 participants per item [32]. Since OVS had 12 items, the sample size of 140 could be adequate, considering a potential dropout rate of approximately 10%. Following recommendations [33] for exploratory factorial analysis (EFA) and confirmatory factorial analysis (CFA) to use not the same participants, the sample size was set to at least 300 in this study. This study was approved by the Ethics Committee of the Lithuanian Sports University (No SNTEK-47).

### 2.2. Instruments

#### 2.2.1. Olympic Game-Related Values

The Olympic Value Scale (OVS) is a self-report 12-item instrument created to measure the perception of values in relation to the Olympic Games [22]. Participants rated the extent to which each of the 12 OVS items could be used to accurately describe the values in relation to the Olympic Games, measured on a 7- point scale from 1 = ‘does not describe the Olympic Games at all’ to 7 = ‘describes the Olympic Games very well’. As mentioned earlier, all 12 items capture three dimensions: appreciation of diversity, friendly relations with others, and achievement in competition. Previous studies by Koenigstorfer and Preuss [22,23] showed acceptable levels of internal consistency of each dimension ranging from 0.88 to 0.91.

#### 2.2.2. Other Variables

The study also included other variables that we expected to be related with students’ perceived values related with the Olympic Games. Specifically, there were three ordinary questions presented to participants regarding their interest in the Olympic Games. Students were asked, “Were you interested in the last Olympic Games?” with five response items ranging from 1–“No, I wasn’t interested in the Olympic Games” to 5–“Yes, I was interested very much (watched them on TV, followed them on the internet, etc.)”. Next, students were asked, “Is it important to you that Lithuania won as many medals as possible in the Olympic Games?” with five response options ranging from 1–“absolutely not important” to 5–“very important”. Participants were also asked, “Are the Olympic Games different from other sports events (e.g., World Championships)?” with five answer items ranging from 1–“not different at all” to 5–“very much different”.

In addition, participants’ attitudes toward fair play in sport were measured by using a 10-item scale [34] which was previously validated in Lithuania [35]. Participants rated each item on a 4-point scale from “strongly agree” to “strongly disagree”. Assessing participants’ attitudes toward fair play, the overall score was calculated with a higher value showing more positive attitudes. Previous studies [35] demonstrated the adequate internal consistency of this scale (0.71).

### 2.3. Procedures

The research procedures included translation procedures, pilot testing, the actual survey, and statistical analysis for validation. The actual study was carried out from November 2019 to May 2020.

The Olympic Value Scale (OVS), which is a 12-item self-report instrument [22], was used for translation. After the permission to use the OVS, it was translated following the recommended procedures for cross-cultural research instrument translation, adaptation, and validation [36,37]. The translation procedure was comprised of several steps. Step 1. Forward translation. Two independent translators translated the OVS into the Lithuanian language. Step 2. Synthesis of the two translated versions. Both translators and a new independent translator together with the research team discussed the translations, their discrepancies, and agreed on the final Lithuanian version. Step 3. Back-translation. Although the scale is made up of single words and this step could be seen as excessive, two independent translators, who were not informed about the concepts explored, translated the scale back into the original language. Step 4. Expert committee. The translated versions were evaluated by three experts. Two of these experts were scientists with experience in research directly related to Olympism and the Olympic Movement. In addition, one of these researchers teaches a module on Olympism and the Olympic Movement to students. The third expert has defended his dissertation on Olympic education but is currently involved in practical activities preparing and coordinating the implementation of Olympic education programs. The final translations into Lithuanian were as follows: nediskriminavimas (anti-discrimination), įvairovė (diversity), lygybė (equality), and tolerancija (tolerance) were used to assess the appreciation of diversity (as an other-oriented value dimension); brolybė (brotherhood), draugystė (friendship), supratimas (understanding), and šilti santykiai su kitais (warm relations with others) were used to assess friendly relations with others (also as an other–centered value dimension); laimėjimai (achievement), geriausių rezultatų demonstravimas (achieving one’s personal best), varžymasis (competition), and pastangų demonstravimas (effort) were used to assess achievement in competition (as a self-centered value dimension). Step 5. Pilot testing. The pilot testing was conducted with 30 students from one university who were native-level speakers of the Lithuanian language. Participants were invited to comment on the clarity of each item and the rating system. Most (95%) students rated the items and the response format as clear. 

The main study was conducted with the university students who studied in programs related to physical education, coaching, sports management, and physiotherapy. Researchers directly contacted the students and invited them to participate in the survey. Prior to administering the questionnaire, all participants were informed that participation was voluntary and that the data collected would remain confidential. After informed consent was obtained, students were instructed on how to complete the questionnaire. All students completed the questionnaires during the class time. The researcher was always present while the subjects completed the questionnaires.

### 2.4. Data Analysis

To assess scale structural validity, both EFA and CFA were performed. Following the recommendation, the total sample of 365 was randomly split into two groups [33]. The first group (*n* = 182) of participants was used for the EFA and the other group (*n* = 183) for the CFA. First, the EFA was performed to synthesize the structure of the relations among the 12-scale items. Next, the CFA was performed to verify the factor structure. To evaluate model fit, χ^2^ statistics, comparative fit index (CFI), Tucker–Lewis index (TLI), normed fit index (NFI), incremental fit index (IFI), root mean square error of approximation (RMSEA), and standardized root mean square residual (SRMR) [38] were examined. A value of 0.90 or higher for CFI, TLI, NFI, and IFI as well as an RMSEA of 0.08 or lower served as indicators for acceptable model fit [39]. Multigroup analyses to investigate gender invariance were carried out. The following types of invariances were considered: configural, metric, scalar, and residual invariance. Further, ∆χ^2^ tests and differences in ∆CFI were analyzed between the constrained and the unconstrained models. Differences in ∆RMSEA were also considered. The invariance criteria used were ∆CFI ≤ 0.01 and ∆RMSEA ≤ 0.015 [40]. Descriptive statistics, skewness and kurtosis, and internal consistency (Cronbach’s alpha coefficient) of each of the factors were also analyzed. Furthermore, convergent validity was evaluated by assessing the composite reliability and average variance extracted (AVE). A composite reliability value of 0.70 or higher was considered adequate. An AVE value higher than 0.5 was considered as acceptable [41]. Discriminant validity was assessed by examining the square root of the AVE. When this value of each measure variable is greater than the correlation coefficient between the variables, it indicates that discriminant validity is established [42]. Correlational analysis was conducted to examine relations between different variables. All analyses were conducted using IMB SPSS Statistics 25 (IBM Corp., Armonk, NY, USA) and AMOS version 24 (IBM Corp., Chicago, IL, USA).

## 3. Results

### 3.1. Participants’ General Characteristics

In this study, 365 valid data of university students (male *n* = 182, female *n* = 183) who were aged between 18 and 31 years (*M* age = 22.02, SD = 6.58) were used for statistical analysis. All participants were native-level speakers of the Lithuanian language. In the sample, 55.3% (72.5% male) participated in sports with a mean of sport experience in their respective/current sport of 8.70 years (SD = 4.32) They participated in 22 different sports, including basketball, football, athletics, martial arts, and other sports (e.g., handball, swimming (Table 1)). Among athletes, 65 were currently or had recently competed at the international level, 93 at the national, and 44 at the local level.

Table 1 shows that among all study participants, less than 20% of them regularly watched the most recent Olympic Games. More than one third agreed that Olympic Games differed from other sports events very much. Additionally, almost 40% of participants stated that they cared more about how many medals the representatives of the national team won at the Olympic Games.

### 3.2. Exploratory Factor Analysis

The factor structure of the 12vitems of the LT-OVS was examined using the EFA with the principal axis factoring as the extraction method. Initially, the Kaiser–Meyer–Olkin (KMO) of sampling adequacy and Bartlett’s test of sphericity were calculated to verify the appropriateness of an EFA. Only items with a strong loading (0.50 or higher) on one factor were retained to form latent variables. Bartlett’s test of sphericity was significant (χ^2^ (66) = 2090.69, *p* < 0.001), and the KMO test yielded a value of 0.84, suggesting that the sample was adequate and sufficiently factorable [31]. Results showed a three-factor emergence with eigenvalues greater than 1 explaining 80.63% of the total variance. Table 2 presents the factor structure. All item loadings on one factor were higher than 0.50. However, one item needs to be noted, “Equality”, as it is cross-loading. Although this item loading on the initial factor was greater than 0.50, we conducted the CFA to test whether the 12-item and subsequently 11-item three-factor models were good.

### 3.3. Confirmatory Factor Analysis

The results of the CFA showed that all indices (Chi-Square Fit Index [χ^2^ (51), *n* = 183) = 137.1, *p* < 0.001]; CFI = 0.96; TLI = 0.95, NFI = 0.94, IFI = 0.96, RMSEA (0.08–0.12) = 0.08) were adequate and suggested data fit to the 12-item LT-OVS structure. Standardized factor loadings are presented in Table 2. Next, we tested an alternative 11-item model. The results of the CFA showed that this alternative model of the OVS also provided an acceptable model fit (Chi-Square Fit Index [χ^2^ (45), *n* = 198) = 125.5, *p* < 0.001]; CFI = 0.97; TLI = 0.97, NFI = 0.96, IFI = 0.97, RMSEA (0.06–0.11) = 0.08). As changes of all indices were minor, it cannot be said that the 11-item model is better than the original 12-item model.

### 3.4. Gender Invariance Analysis

Aiming to evaluate whether the scale structure was invariant to gender, multigroup analysis was conducted. As seen in Table 3, models suggest support for invariance between male and female, showing that the same number of factors was present in both groups and remained associated with the same items (configural invariance), the LT-OVS factors had a similar understanding in both groups (measurement invariance), and latent and observable means were valid in both groups when means were compared (scale invariance). It was found that after testing, residual invariance was not met as the ∆CFI cut-off value was higher than >0.010.

### 3.5. Descriptive, Reliability and Validity Analysis

Furthermore, we assessed the internal consistency of each of the LT-OVS subscales and the convergent and discriminant validity. Table 4 provides the descriptive statistics, the Cronbach’s alpha, composite reliability, average variance extracted (AVE), and square root of the AVE, as well as the correlations between the factors. It was found that the internal consistency of all the factors was higher than 0.70, ranging from 0.80 to 0.95, which shows that the measures were internally consistent. As shown in Table 4, the composite reliability values were above 0.80 for all the factors. Moreover, the convergent validity criterion was also met as AVE was above 0.50. The criterion of discriminant validity was also met as correlations between the LT-OVS factors were lower than the respective square root of the AVE values (>0.80).

Next, we examined the associations between the LT-OVS factors and other variables. Specifically, we first tested the relationship between the LT-OVS factors and participants’ attitudes towards fair play in sport. We expected that those with more positive attitudes towards fair play would exhibit a higher value on appreciation of diversity and friendly relations with others. Our expectations were met, showing that attitudes towards fair play showed stronger correlation with the factor of the appreciation of diversity (r = 0.20, *p* < 0.01) than that of friendly relations with others (r = 0.11, *p* < 0.05) (Table 5). A statistically significant correlation between attitudes toward fair play and achievement in competition was not found. We also assessed relations between the perceived values and variables related with the Olympic Games. Table 5 shows that seeing the differences between the Olympic Games and other sports events was positively related to all three perceived variables of the Olympic Values. Interest in the Olympic Games was also positively related to perceived participants’ friendly relations and achievements in competitions. Finally, it was found that those to whom the achievements of the country’s athletes at the Olympic Games were of greater importance emphasize such Olympic Games-related values as friendly relations and achievements in competitions.

## 4. Discussion

The fundamental principles of Olympism imply that the Olympic Games should be based on Olympic spirit and promote such values as friendship, solidarity, and fair play [1]. However, some trends show a growing skepticism among the population regarding the Olympic Games, especially in Europe [20,43], which encourages deep interest in the dissemination of the Olympic Values. The recently developed OVS [22] could be a useful research instrument in better understanding how people perceive values related with the Olympic Games. Aiming to verify to what extent this scale can be used in other countries, the purpose of this study was to adapt and validate the Lithuanian version of the OVS.

Results showed that the LT-OVS could be used in future studies involving Lithuanian-speaking people and also provided support for the need for future validation of this scale in other countries. However, several key aspects that emerged during the scale adaptation need to be discussed. Koenigstorfer and Preuss [22,23] confirmed that the OVS had three factors—namely, appreciation of diversity, friendly relations with others, and achievement in competition. We also expected that the factor structure of the LT-OVS in a Lithuanian student sample would reflect the original three-factor structure, and our study confirmed the multidimensional structure of this scale. However, while examining the structure, it was observed that the statement “Equality” cross-loaded on the LT-OVS dimensions of appreciation of diversity and friendly relations with others. So, two steps were taken. We examined the factorial structure of the LT-OVS with 12 items and additionally with 11 items. The CFA results confirmed that the shorter version with the elimination of the item “Equality” was not better than the original 12-item scale. In fact, in previous studies, Parry [44] linked equality to anti-discrimination, which had a similar meaning to appreciation of diversity, as was confirmed in Koenigstorfer and Preuss’ [22] study. Thus, it can be stated that both theoretically and empirically, it is more expedient to use the 12-item LT-OVS. 

Our results supported the gender measurement invariance of the LT-OVS. Specifically, the multi-group analysis showed that configural, measurement, and scale invariance criteria were meet [40]. However, residual invariance was not met, but some authors noted that in social science research, this parameter is optional [45]. So, this may not implicate the absence of the LT-OVS measurement invariance.

We found high Cronbach’s alpha coefficients for the three subscales of the LT-OVS (ranging from 0.80 to 0.95), which was in line with those seen in a study with samples from the USA and Germany [22], as well as Brazil [23]. These results indicated an acceptable to excellent level of internal consistency for the Lithuanian version. 

In this study, we also analyzed correlations among the LT-OVS dimensions and attitudes towards fair play in sport as well as attitudes towards the Olympic Games. It can be summarized that students who emphasized values of appreciation of diversity and friendly relations with others as more related with the Olympic Games had more positive attitudes toward fair play, and this was what we expected. However, the dimension of achievement in competition was not related to attitudes toward fair play. Based on the theory of self-serving behavior [46], Koenigstorfer and Preuss [22] identified achievement in competition as a self-centered value dimension, while the other two dimensions are other-centered value dimensions. Therefore, perceiving the Olympic Games more as serving achievement in competition, it can be assumed that participating in them is more about demonstrating one’s superiority over others and winning [22]. Moreover, our results showed that the research participants most associated the Olympic Games with the values of achievement in competition. Meanwhile, fair play was more central for others as it was linked with respect not only for the rules of the game but also respect for others and for the game as such [47]. Therefore, we can say that the established links between the Olympic Games value dimensions and attitudes toward fair play add evidence to the convergent validity of the LT-OVS.

The results also revealed that perceived values related to the Olympic Games were also related with attitudes toward the Olympic Games. Specifically, those believing that the Olympic Games stood for appreciation of diversity, friendly relations with others, and achievement in competition (as perceived values) perceived the Olympic Games as more different from other sports events. Interest in the Olympic Games and the achievements of athletes in them was also positively correlated with the friendly relations with others and achievement in competition dimensions, which is partly congruent with Koenigstorfer and Preuss’ [22] results. However, interest in the Olympic Games and the achievements of athletes were not related to the perception of the Olympic Games as standing for appreciation of diversity. This is partly unexpected because respect for other people—tolerance for people regardless of their background—is one of the fundamental Olympic principles [1]. First, we can assume that the interest of people in the Olympic Games, and at the same time the participation of the country’s athletes and their achievements in them, are more related to personal satisfaction and national pride, as it was found in other studies [16], but not opportunities to promote non-discrimination and tolerance for others in the Games. Second, we might speculate that such results can also be related to some tendencies of tolerance towards people with different backgrounds, for example, attitudes towards migrants [48]. 

Finally, we should mention some methodological implications. Results on validity and reliability confirmed that the OVS could be successfully adapted in other non-Western countries. However, it is necessary to make it clear that this three-factor scale was adapted to the Lithuanian population in a study with university students, as originally it was developed with participants of various ages. Thus, we can expect that the OVS could be easily adapted to other cultural contexts. However, future research needs to take into account the age of participants by adapting this scale to other cultures. Additionally, in the future, studies in Lithuania with participants of other ages should re-evaluate the structure of the LT-OVS. 

Several practical implications need to be mentioned. Most of the previous studies were focused on how residents perceived the Olympic Games when their city or country of residence had hosted, would host, or was a candidate for hosting the Games. A recent study by Koenigstorfer and Preuss [23] also examined the relationship between values related to the Olympic Games and peoples’ attitudes towards them, and the intended support of hosting the Olympic Games in their country. However, we are lacking studies on how the values of the Olympic Games are perceived by people from countries not hosting and not candidates to host the Olympic Games. Such studies are important as the Olympic Games should be beneficial not only for the hosting city or country, but also for the whole society by promoting Olympic ideals. Understanding how people perceive values related to the Olympic Games can show what they believe about what stands behind the Olympic Games and how the core values of Olympism are disseminated in the population. Results on the dimensions of the Olympic Games may be useful for the National Olympic Committees, Olympic Academies, and for educational purposes at schools, especially for the Olympic education programs.

Our study is not without limitations. Participants of our study were university students. It should also be mentioned that half of the study participants were students actively engaged in sports and, in general, respondents were from study programs related to sport and physical activity. Although the results of the OVS validation in the Lithuanian language are promising, the LT-OVS may only reflect how youth perceive values related with the Olympic Games. Therefore, it is still not clear whether this three-factor scale would be supported by studies with older participants. Secondly, we related the LT-OVS factors with variables that related to attitudes and perception, but we did not include actual behavioral variables. Third, we did not evaluate the temporary stability of the scale. 

## 5. Conclusions

The current study has provided evidence that the LT-OVS is a reliable and valid instrument to be used in the Lithuanian context to assess how people perceive values in relation to the Olympic Games. However, the Lithuanian version of the OVS should be further investigated with more diverse and more extensive populations, involving more wide-ranging age groups.

## Figures and Tables

**Table 1 ijerph-19-04360-t001:** Study sample description.

		Total (*n* = 365)	Male (*n* = 182)	Female (*n* = 183)
Age M(SD)		22.02 (6.58)	22.04 (6.76)	22.01 (6.41)
Participation in sport % (*n*)	Yes	55.3 (202)	72.5 (132)	38.3 (70)
	No	44.7 (163)	27.5 (50)	63.7 (113)
Sport experience M(SD)		8.74 (4.29)	8.94 (4.53)	8.35 (3.83)
Type of sport % (*n*)	Individual	43.1 (87)	32.6 (43)	62.9 (44)
	Team	41.6 (84)	50.0 (66)	25.7 (18)
	Martial	15.3 (31)	17.4 (23)	11.4 (8)
Level of competition % (*n*)	International	32.2 (65)	28.8 (38)	38.6 (27)
	National	46.0 (93)	49.2 (65)	40.0 (28)
	Local	21.8 (44)	22.0 (29)	21.8 (44)
Participants who were regularly interested (watched on TV, followed on the Internet etc.) in the most recent Olympic Games % (*n*)		16.7 (61)	20.9 (38)	12.6 (23)
Participants who perceived Olympic Games as very different from other sports events (i.e., World Championship) % (*n*)		38.1 (139)	45.1 (82)	31.1 (57)
Participants who cared very much how many medals the National Olympic Team would win % (*n*)		41.1 (150)	44.0 (80)	38.3 (70)

**Table 2 ijerph-19-04360-t002:** Factor loadings and communalities (h^2^) for the LT-OVS in the EFA (*n* = 182) and in the CFA (*n* = 183).

Factors Corresponding Items	EFA			*h^2^*	CFA Factor Loading
	Factor Loadings				
	1	2	3		
1. Friendly relations with others					
Friendship	**0.94**	0.17	0.26	0.98	0.99
Brotherhood	**0.90**	0.15	0.18	0.87	0.90
Warm relation with others	**0.90**	0.13	0.23	0.88	0.91
Understanding	**0.85**	0.25	0.17	0.81	0.86
3. Appreciation of diversity					
Anti-discrimination	0.14	0.17	**0.77**	0.65	0.61
Tolerance	0.34	0.28	**0.78**	0.81	0.87
Diversity	0.17	0.37	**0.66**	0.70	0.74
Equality	0.48	0.18	**0.64**	0.68	0.79
2. Achievement in competition					
Achieving one’s personal best	0.24	**0.87**	0.16	0.72	0.88
Competition	0.15	**0.90**	0.22	0.88	0.93
Effort	0.15	**0.90**	0.18	0.86	0.91
Achievement	0.24	**0.78**	0.21	0.84	0.79

Note: Loadings >0.40 are shown in bold.

**Table 3 ijerph-19-04360-t003:** Fit indices for the measurement invariance of the LT-OVS across gender.

Models	χ^2^	df	∆χ^2^	∆df	CFI	∆CFI	RMSEA	∆RMSEA
Male	143.81 **	51			0.955		0.078	
Female	139.09 **	51			0.942		0.079	
Configural invariance	199.50 **	102			0.957		0.073	
Measurement invariance	221.11 **	111	21.6	9	0.951	0.006	0.074	0.001
Scale invariance	239.31 ***	117	39.81	15	0.946	0.011	0.076	0.003
Residual invariance	296.25 ***	129	96.75	27	0.926	0.031	0.085	0.012

Note: ** *p* < 0.01; *** *p* < 0.001.

**Table 4 ijerph-19-04360-t004:** Descriptive statistics, internal consistency, and validity analysis of the variables.

Factors	M (SD)	Skewness/Kurtosis	Cronbach’s α	AVE	Composite Reliability	1	2	3
1. Friendly relations with others	5.34 (1.40)	−0.69/0.12	0.95	0.84	0.95	*0.91*		
2. Appreciation of diversity	5.65 (1.14)	−0.94/1.41	0.80	0.58	0.84	0.60 **	*0.76*	
3. Achievement in competition	6.53 (0.96)	−0.14/0.44	0.92	0.77	0.93	0.37 **	0.51 **	*0.88*

Note: AVE stands for average variance extract. The square root of the AVE is shown in the diagonal (italics). ** *p* < 0.01.

**Table 5 ijerph-19-04360-t005:** Correlation of the values related with Olympic Games with other variables.

Factors	Attitudes toward Fair Play	Interest in Olympic Games	Perceived Differences of Olympic Games	Achievements of the National Team in Olympic Games
1. Friendly relations with others	0.11 *	0.18 **	0.15 **	0.11 *
2. Appreciation of diversity	0.20 **	0.08	0.14 **	0.04
3. Achievement in competition	0.10 *	0.11 *	0.17 **	0.11 *

Note: * *p* < 0.05, ** *p* < 0.01.

## Data Availability

The data presented in this study are available on request from the corresponding author.

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
