# Peer review of "Adaptation and Validation of the Lithuanian-Language Version of the Olympic Value Scale (OVS)"

_ijerph, 2022, doi:10.3390/ijerph19074360_

Round 1

Reviewer 1 Report

Sukys and colleagues in their article examined the reliability and validity of the Lithuanian Version of the Olympic Value Scale (OVS). Overall, the authors have done a good job. Nevertheless, some specific comments are provided below.

Abstract:

- Please add in more specific results.

Introduction:

- L 90 Make “A significant step was taken...” a new paragraph.

- L 73 Add abbreviation next to International Olympic Committee since you use the abbreviation in the future.

- L 91-93 Add in more explanation of the subscales of the OVS.

- I believe it is important to create valid and reliable questionnaires in other languages, however, add in a little more information on why specifically it important to create a Lithuanian version.  

Materials and Methods

- L 130 Add more on how these individuals are considered experts.

- L 169-177 You might consider moving the additional sample description questions with explanation earlier. I was unclear what these were in the table until later in the manuscript.

- It sounds like the questionnaire was distributed in person as stated with L190-191. Add in additional information on how this was exactly done. Were they collected during a class period? Individually?

- L 199 Make sure all abbreviations are spelled out the first time.

Results

- The authors did a good job in data treatment. Tables are helpful to understand the analyzes.

- Perhaps add in a little more information somewhere earlier in the manuscript on how these additional variables of fair play and perceived values related with Olympic Games may be used with the OVS.

Discussion

- The discussion is clear in explaining the results and usefulness of the instrument.

Limitations:

- I suggest adding in how being an athlete or former athlete may have impacted the results in terms of sex differences and level of competition. Also, how does studying physical education, coaching, sports management and physiotherapy impact the results?

I suggest the authors edit the manuscript again for small grammatical errors.

Reviewer 2 Report

The present study presents information in a clear and objective way. In addition, the authors follow the necessary steps for the presentation of a translation and validation study of the questionnaire. I believe that the study guarantees the reproducibility of the questionnaire.   Regarding specific questions in the text: Introduction:   The 4th paragraph is too long, it can be divided into two other paragraphs to make reading less tiring.   Materials and methods:   In section 2.3 Participants, the authors do not present the eligibility criteria of the sample selected for the study.   Item 2.3 is repeated three times, in the sections: 2.3 Participants; 2.3 Instruments; 2.3 Procedures. It is necessary to make this correction by adapting the text to the correct sequence.   The results and discussion sections do not require major corrections.

Reviewer 3 Report

This is an interesting paper. In many cases the authors have made a diligent attempt to address the comments I raised in relation to the previous version, and I think this has resulted in a significantly strengthened manuscript. In a few cases the authors have indicated their disagreement with my comments, and, while I continue to hold my original views that those changes would have made the paper better, I nonetheless think the paper is now in a state good enough to publish. I think this paper can make a useful contribution to the IJERPH journal!
